# Combined orbital tomography study of multi-configurational molecular adsorbate systems

Pavel Kliuiev [1,4], Giovanni Zamborlini [2,5], Matteo Jugovac [2], Yeliz Gurdal[3,6], Karin von Arx[1], Kay Waltar[1], Stephan Schnidrig[3], Roger Alberto[3], Marcella Iannuzzi [3], Vitaliy Feyer [2], Matthias Hengsberger [1], Jürg Osterwalder [1] & Luca Castiglioni [1]*

Molecular reactivity is determined by the energy levels and spatial extent of the frontier orbitals. Orbital tomography based on angle-resolved photoelectron spectroscopy is an elegant method to study the electronic structure of organic adsorbates, however, it is conventionally restricted to systems with one single rotational domain. In this work, we extend orbital tomography to systems with multiple rotational domains. We characterise the hydrogen evolution catalyst Co-pyrphyrin on an Ag(110) substrate and compare it with the empty pyrphyrin ligand. In combination with low-energy electron diffraction and DFT simulations, we fully determine adsorption geometry and both energetics and spatial distributions of the valence electronic states. We find two states close to the Fermi level in Co-pyrphyrin with Co $3d$ character that are not present in the empty ligand. In addition, we identify several energetically nearly equivalent adsorption geometries that are important for the understanding of the electronic structure. The ability to disentangle and fully elucidate multi-configurational systems renders orbital tomography much more useful to study realistic catalytic systems.

[1] Department of Physics, University of Zurich, 8057 Zurich, Switzerland. [2] Peter Grünberg Institute (PGI-6), Forschungszentrum Jülich, 52425 Jülich, Germany. [3] Department of Chemistry, University of Zurich, 8057 Zurich, Switzerland. [4] Present address: Department of Applied Physics, Aalto University School of Science, FI-00076 Aalto, Finland. [5] Present address: Technische Universität Dortmund, Experimentelle Physik VI, 44227 Dortmund, Germany. [6] Present address: Department of Bioengineering, Adana Alparslan Turkes Science and Technology University, 01250 Adana, Turkey. *email: castiglioni@physik.uzh.ch

The frontier orbitals largely determine electronic and optical properties of molecules and their reactivity. Upon adsorption of molecules on metallic or dielectric surfaces, the electronic structure can significantly change due to interaction with the substrate such as charge transfer[1,2], hybridisation[3] or even covalent bonding (chemisorption)[4]. As a consequence, both geometry and electronic structure of the adsorbed molecules change, resulting in shifts of the electronic levels or even reordering of the orbital hierarchy. Modern electronic structure theory has evolved into a powerful method to calculate molecular wave functions and predict chemical properties, yet the theoretical description of adsorbed systems remains challenging. In order to understand or predict the reactivity and properties of adsorbed molecules, experimental access to their electronic structure is highly desirable. Scanning probe methods such as scanning tunneling microscopy (STM) can in principle image the charge distribution with submolecular resolution[5,6], but give no direct access to the electronic wave function and are typically restricted to the highest occupied and lowest unoccupied molecular orbitals (HOMO and LUMO, respectively).

Angle-resolved photoelectron spectroscopy (ARPES)[7] enables measurement of the electron momenta since the parallel component is conserved in the photoemission process. Orbital tomography based on ARPES data takes advantage of the fact that the electron momentum distribution and the real space orbital density can be directly related via Fourier transform under certain conditions[8], most importantly the description of the photoemission final state as a free electron (i.e. a plane wave). Both validity of the plane-wave approximation[9–11] and the quantum mechanical interpretation[12] have been discussed in the past. The method was successfully used for the reconstruction of molecular orbitals of polycyclic aromatic molecules solely from ARPES data[8,13–15]. The reconstruction via iterative phase retrieval algorithms[16] requires two preconditions to be met: (i) the experimental data must be oversampled[17] and (ii) all molecules must be oriented in the same way. While (i) can easily be fulfilled by choice of the right experimental conditions (i.e. primarily sufficient momentum resolution), the orientation of the molecule on the substrate (ii) is dictated by aforementioned substrate–molecule interactions. The requirement of single orientation is thus rather limiting, with only a few selected systems having been studied so far[8,13,18]. The orientation of molecules with respect to the substrate, i.e. the so-called molecular registry, can conventionally be determined by STM[19] or x-ray photoelectron diffraction[20,21].

In this work, we demonstrate that a combination of orbital tomography, low-energy electron diffraction (LEED) and density functional theory (DFT) can address the combined structural and electronic aspects, including orbital energies, their correct hierarchy and the molecular wave functions. We chose the complex macrocycle cobalt-pyrphyrin (CoPyr) to demonstrate viability, effectiveness and generality of our approach. CoPyr is a promising water reduction catalyst to produce hydrogen in photochemical water splitting[22,23]. The pyrphyrin ligand (Pyr) was first synthesised by Ogawa[24] and is currently employed as carrier for catalytically active transition metals[25]. Both CoPyr and Pyr adsorb on the Ag(110) surface in multiple rotational domains because the high-symmetry axes of the molecules are not aligned with the high-symmetry direction of the (110) substrate. Yet using our combined approach and comparing the experimental photoelectron momentum maps (PMMs) with DFT simulations, we can fully determine the molecular registries. In addition, we can identify five valence molecular states in CoPyr. The two highest levels have contributions from the Co 3d orbitals and are not present in Pyr. Importantly, the found orbital hierarchy differs from gas-phase DFT calculations and underlines the importance of high-level simulations and experimental data.

## Results and discussion

**Adsorption geometries.** DFT simulations were performed to identify possible adsorption geometries (AGs) of CoPyr and Pyr on Ag(110). The three lowest energy configurations are shown in Fig. 1. The additional interaction of the Co central atom with the substrate leads to significantly stronger adsorption of CoPyr compared to the free Pyr ligand. For both CoPyr and Pyr on Ag(110), the most stable configuration is AG 3, where the angle between the CN-groups of CoPyr and Pyr and the $[1\bar{1}0]$ high symmetry direction of the substrate is $\zeta = -39°$ and $\zeta = -38°$, respectively. In the other two cases, the molecules are oriented parallel (AG 2) and perpendicular (AG 1) with respect to the $[1\bar{1}0]$ direction, and their adsorption energies are higher. We note though the small difference between AGs 2 and 3 of CoPyr of only 30 meV, thus potentially facilitating the adsorption of CoPyr molecules in both configurations.

In a next step, the surface lattices of the CoPyr and Pyr adlayers were determined by LEED. Figure 2 shows experimental LEED patterns of CoPyr and Pyr on Ag(110) together with the simulated patterns done in the framework of a geometric LEED theory. The simulations suggest that CoPyr and Pyr molecules are ordered in two surface lattices, which are mirror counterparts of each other with respect to the $[1\bar{1}0]$ high symmetry direction of the Ag substrate. In Fig. 2a, e, these lattices are shown in blue and red and in the following will be referred to as lattice 1 and lattice 2, respectively.

The LEED analysis along with the AGs inferred from the DFT simulations allows for a visualisation of possible spatial orientations of CoPyr and Pyr on the Ag(110) surface. We notice that the lattice constants of lattices 1 and 2 are on the order of the size of the molecule ($c_1 \approx c_2 = 12.5$ Å, distance between CN groups in CoPyr: 11.8 Å). The simultaneous existence of both lattices over the spatial extent of a few lattice constants therefore appears unlikely. Hence, we conclude that in the case of CoPyr on Ag(110) lattices 1 and 2 exist separately in the form of domains. For AG 3 with $\zeta = -39°$ there is a mirrored AG 3′ with $\zeta = 39°$. The other two AGs (1 and 2) are symmetric with respect to the $[1\bar{1}0]$ direction and thus there are no mirror configurations. In terms of electrostatic repulsion, the CN moieties are the most important features in Pyr and CoPyr. Consequently, we look for combinations of AGs and lattices with maximum distance between the CN moieties of neighbouring molecules. In lattice 2, neighbouring CN groups are in close proximity in AG 1 (Fig. 2b), whereas they are further apart in both AG 2 and AG 3 (Fig. 2c, d). The molecular registries in lattice 1 can be obtained by mirroring the correspondent lattice points and AGs with respect to the $[1\bar{1}0]$ direction of the substrate (see Supplementary Fig. 1). The same procedure can be applied to determine possible molecular registries for Pyr on Ag(110). For symmetry reasons there also exist mirror-symmetric configurations for AG 2 and AG 3 (see Supplementary Fig. 2). In all shown configurations (Fig. 2f–h), the distance between neighbouring CN groups is reasonably large, and thus none of these configurations can be excluded a priori.

By combining AGs obtained from the DFT simulations with lattices derived from LEED, we are able to predict possible combinations of molecular registries of CoPyr and Pyr molecules. We find that in principle, multiple rotational domains of the CoPyr and Pyr molecules can exist on the Ag(110) surface. However, final conclusions can be made only after comparing the ARPES data with the DFT simulations.

**ARPES data and electronic structure.** Silver is a suitable substrate for studies of molecular adsorbates because it has a low density of states between the Fermi energy and the onset of the d-band at around 4.0 eV binding energy[26]. We could locate five

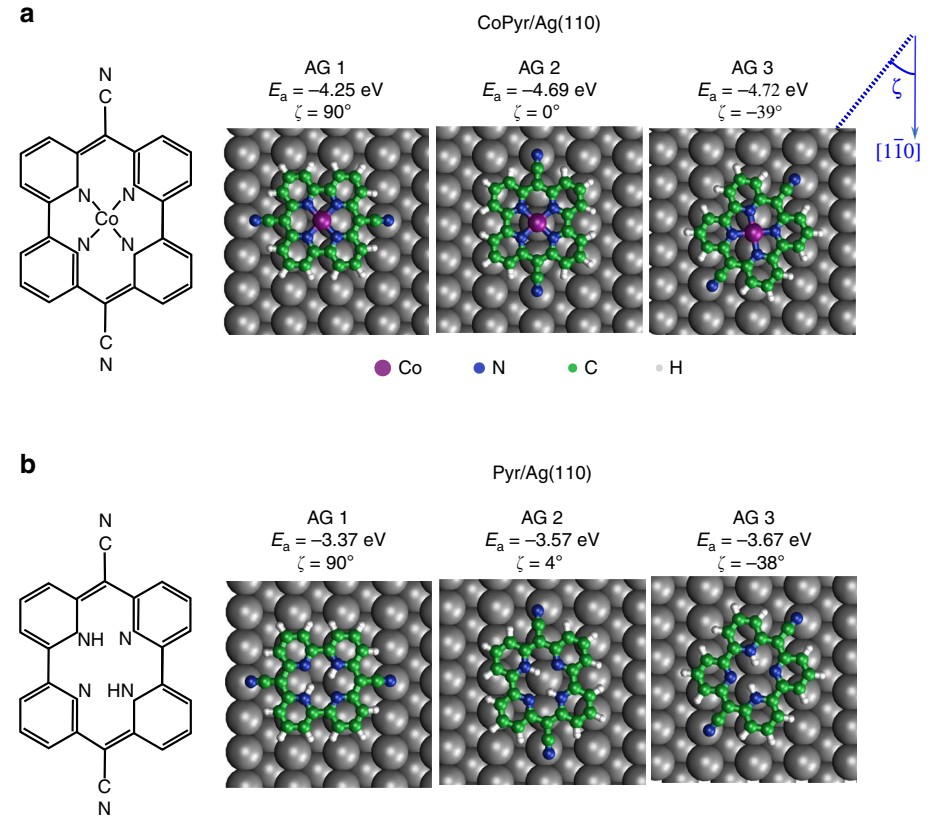

**Fig. 1** Adsorption geometries (AGs) and corresponding adsorption energies ($E_a$) on the Ag(110) surface. **a** Co-pyrphyrin (CoPyr) on Ag(110). **b** Pyrphyrin (Pyr) on Ag(110). The angle between the molecular symmetry axis (linking the two CN groups) and the Ag substrate [1̄10] direction is indicated by $\zeta$

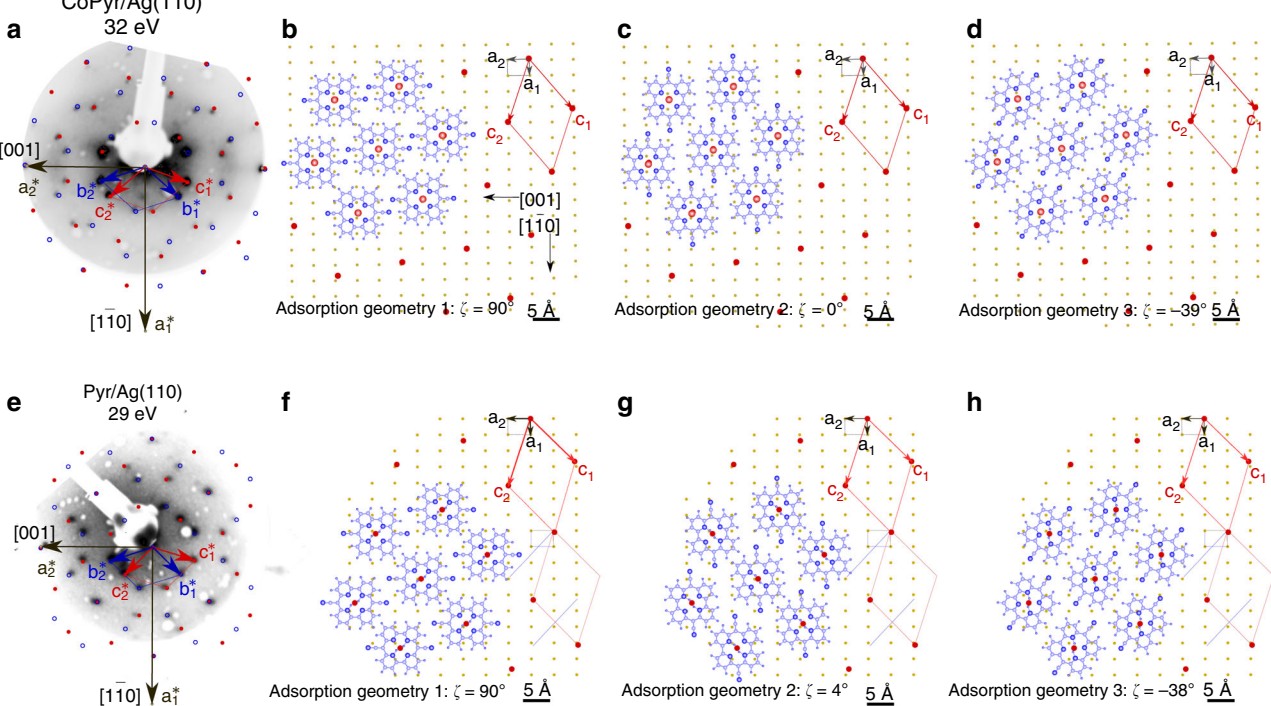

**Fig. 2** Analysis and simulation of low-energy electron diffraction (LEED) data. **a–d** Co-pyrphyrin (CoPyr) on Ag(110). **a** LEED pattern acquired at 32 eV beam energy. Overlay: simulated reciprocal lattices of molecular adlayers (lattice 1 (blue), lattice 2 (red)). **b–d** Visualisation of possible CoPyr molecular registries on Ag(110) in lattice 2. **e–h** As in **a–d**, but for pyrphyrin (Pyr) on Ag(110)

molecular states in case of CoPyr and three in case of Pyr within this valence region (Fig. 3). The lowest observed states (*cp.i* and *p.i*) do not appear as strong peaks in the angle-integrated data but can be clearly distinguished in the momentum maps as will be seen later. The three lowest-energy states have nearly the same binding energy in CoPyr and Pyr. The two highest states in CoPyr (*cp.iv* and *cp.v*) are missing in Pyr. This is also reflected in the displayed projected density of states (PDOS, projection on molecular C, N and Co atoms) from the DFT calculations of the different AGs.

The PMMs for each of the observed molecular states are shown in Fig. 4 together with various simulations. The raw images (Fig. 4a, f) are characterised by a number of blobs, i.e. intensity maxima, of different shape, size and mutual location, found between 1 and 2 Å$^{-1}$ from the centre, which originate from

molecular states. Sharp, crescent-like features are due to emission from the sp-band of Ag(110). To facilitate comparison with the simulations, the Ag(110) substrate background was subtracted from the raw PMM and they were normalised with respect to the light polarisation, i.e. the $|\mathbf{A} \cdot \mathbf{k}_f|^2$ factor (Fig. 4b, g), see Supplementary Methods and Supplementary Fig. 3).

Within the plane-wave final-state approximation, the PMM can be computed by Fourier transform of the DFT-computed molecular orbitals[8,15], multiplication with an instrument transfer function and projection of the resulting momentum distribution to the corresponding 2D map (refer to Supplementary Methods for details). To highlight the robustness of the molecular wave functions, we use simulated PMMs based on the simple gas-phase DFT calculations (Fig. 4c, h), high-level calculations are shown in the Supplementary Fig. 4). First, we compare PMMs in all three

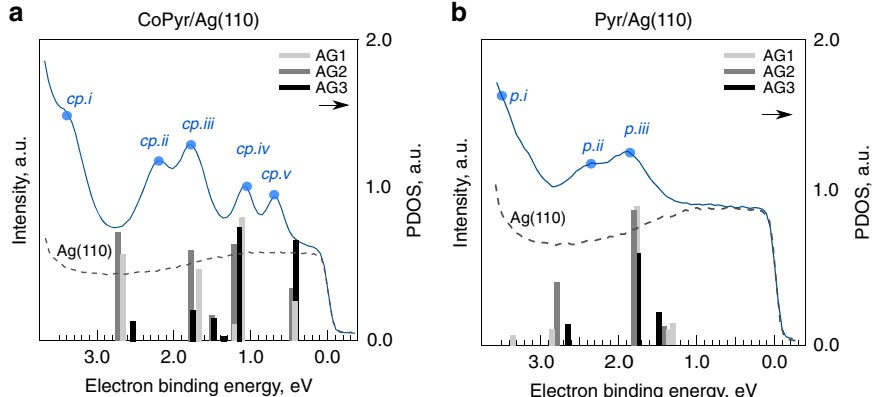

**Fig. 3** Valence region photoelectron spectra. **a** Co-pyrphyrin (CoPyr) on Ag(110). **b** Pyrphyrin (Pyr) on Ag(110). CoPyr molecular states are marked as *cp.i–cp.v*, those of Pyr as *p.i–p.iii*. The projected density of states (PDOS) obtained from DFT for indicated adsorption geometries (AGs) is represented by grey-shaded bars. The Ag(110) substrate background is indicated by the grey dashed line

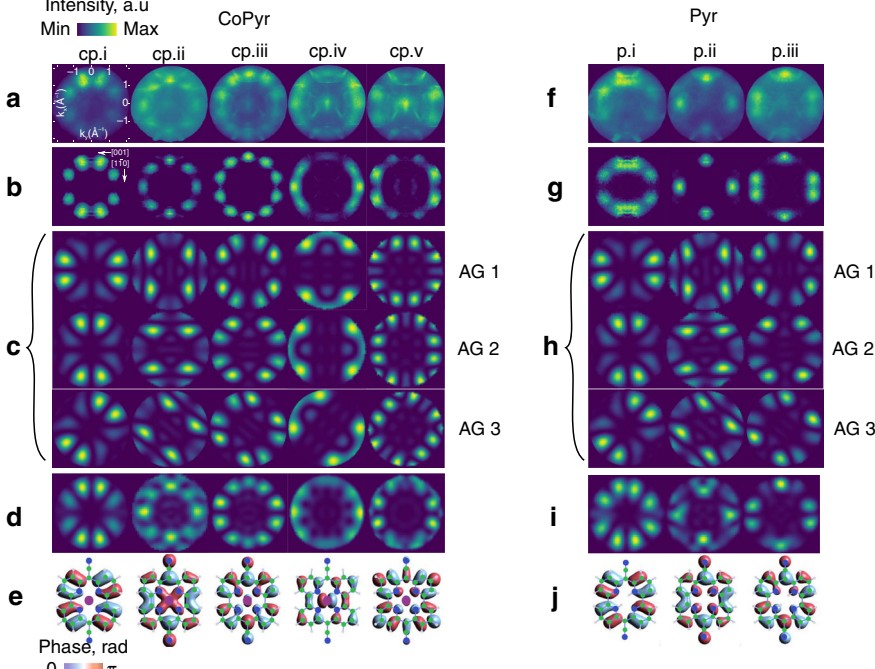

**Fig. 4** Angle-resolved photoelectron spectroscopy (ARPES) data and simulated photoelectron momentum maps (PMMs). **a–e** Co-pyrphyrin (CoPyr). **f–j** Pyrphyrin (Pyr). **a, f** Raw ARPES data on Ag(110) substrate. **b, g** PMMs after background subtraction and normalisation. **c, h** Simulated PMMs for indicated adsorption geometries (AG) based on gas-phase DFT data. **d, i** Incoherent sum of PMMs: CoPyr (AGs 2, 3, 3′ weighted 1.5:1:1), Pyr (AGs 3, 3′ weighted 1:1). **e, j** Corresponding molecular orbitals

AGs to the experimental data. The simulated PMMs reveal the presence of very similar intensity distributions in momentum space that differ mostly by their azimuthal orientation, in accordance with the orientation of the molecule on the substrate. We find that no single AG alone can reproduce the experimental data well. Our previous LEED analysis suggested the existence of multiple rotational domains. Thus, to verify this possibility, we performed a detailed analysis by combining the PMMs from individual AGs and computing their incoherent sums. In case of CoPyr, summing up the PMMs of AGs 2, 3 and 3′ weighted by 1.5:1:1 (Fig. 4d) gives best agreement with the experimental data for all five measured molecular orbitals (Fig. 4b). Thus, we conclude that CoPyr molecules on Ag(110) are adsorbed in three different AGs with the CN groups oriented at 0°, 39° and −39° with respect to the [1$\bar{1}$0] high symmetry direction. This result correlates well with the molecular registries deduced from the LEED analysis (Fig. 2b–d) and it is in line with the computed adsorption energies, which are nearly the same for AGs 2 and 3 and considerably lower than for AG 1. The same procedure was applied to the Pyr data. All three AGs (Fig. 4h) reveal similar intensity distributions in momentum space, but again no single AG can reproduce the experimental data. The best match with the experiment was found with a combination of AGs 3 and 3′ weighted 1:1. This AG shows the largest separation of neighbouring CN groups (Fig. 2h) and has the lowest computed adsorption energy.

While the ARPES data are indeed very helpful for the determination of the AG, they provide much further insight into the electronic structure. As suspected from the binding energies in the photoelectron spectra, the PMMs now irrevocably confirm the equivalence of the three lowest observed valence states in CoPyr and Pyr. The two highest states in CoPyr (cp.iv and cp.v) are characterised by combinations of Co 3d wave functions with the ligand's π-system. Upon careful evaluation of the ARPES data, we could not find any indication of population of the Co $3d_{z^2}$ orbital. This is an important finding since electrons are most likely transferred from this orbital to the water in the catalytic process. Since CoPyr is supposed to work as a photochemical system, the Co $3d_{z^2}$ orbital is populated by photoexcitation and must thus be empty in thermal equilibrium.

We note that the experimentally determined orbital hierarchy strongly differs from the gas-phase DFT prediction where the HOMO corresponds to cp.iii for example (refer to Supplementary Tables 1 and 2 for details). The higher-level calculations of the actual adsorbate systems can in principle reproduce the right hierarchy but the identification is greatly facilitated by the ARPES data. Only the PMM fingerprint of each molecular state enables an unambiguous identification of the corresponding molecular orbital. Low-level DFT computations are thus problematic for the prediction of the reactivity and electronic properties of such systems since they are unable to identify the actual frontier orbitals. In view of the potential use of CoPyr as water reduction catalysts, it is clear that such ARPES data must serve as benchmark for higher level DFT computations.

Our results have direct implications for the reconstruction of real space molecular orbital distributions with phase retrieval algorithms. For iterative phase retrieval algorithms[16,27,28], it plays no role if the features in the experimental PMMs are due to photoemission from a single or multiple rotational domains of the molecules. Solving the phase problem, whose solution was shown to be unique in two dimensions[29], one will always be able to reconstruct some amplitude and phase distributions in real space, provided the oversampling conditions are fulfilled[17]. Thus, to the best of our knowledge, unambiguous interpretation of 2D orbital distributions reconstructed from experimental photoemission data is possible only if the photoemission data originate from a single-orientational domain of molecules[8,13–15]. Presently, we cannot give any meaningful physical interpretation for real space distributions reconstructed from an incoherent superposition of intensity distributions in reciprocal space. Thus, for the purposes of orbital tomography, aiming at reconstruction of real space 2D orbital distributions, the origin of the experimental data must always be clarified.

**Conclusion.** In this study, we show a systematic analysis of LEED and ARPES data combined with DFT calculations that allows for the complete determination of AGs of molecular adsorbates. We studied the water reduction catalyst CoPyr and the empty Pyr ligand on the Ag(110) surface and found that the molecules are ordered in two lattices and adsorbed in multiple AGs. The existence of multiple AGs prohibits the direct reconstruction of molecular orbitals from the ARPES data. The photoemission momentum maps though serve as distinctive fingerprint to link the experimental data with the DFT simulations. The calculation of precise orbital energies is demanding for large and complex molecules. The shape of the wave function itself though turns out to be very robust and largely independent of the level of theory. Since the wave function can be directly linked to the ARPES data via Fourier transform, the experimentally observed states can easily be identified and unambiguously assigned to the DFT molecular orbitals. In that sense, we extend orbital tomography to complex systems with multiple rotational domains and make the method applicable for many molecular interfaces that are relevant to catalysis, organic electronics or photovoltaic devices for example.

## Methods

**Sample preparation and characterisation.** CoPyr and Pyr were synthesised and purified according to literature procedures[23,24]. The surface of Ag(110) was prepared in ultra-high vacuum by alternating cycles of Ar$^+$ sputtering and annealing following standard procedures[30]. Highly ordered adsorbate films on Ag(110) were grown by effusing Pyr molecules at 570 K and CoPyr at 600 K from home-built Knudsen cell-type evaporators (∼40 min. for a monolayer). Substrate cleanliness and quality and thickness of the molecular films were verified by LEED and x-ray photoelectron spectroscopy (XPS).

**Angle-resolved photoemission measurements.** ARPES data were acquired at the NanoESCA beamline of the Elettra synchrotron (Trieste, Italy)[31]. PMMs were recorded in steps of 50 meV kinetic energy using a photoemission electron microscope in the momentum mode[15]. We used p-polarised light with a photon energy of 35 eV impinging the sample under a grazing angle of 25°. Photoelectron spectra were obtained by integrating the PMMs over the complete field of view in momentum space at each kinetic energy. Data for background subtraction were measured with a clean Ag(110) sample at the respective binding energies of the Pyr and CoPyr valence molecular levels. This substrate background was subtracted from the PMMs of Pyr/Ag(110) and CoPyr/Ag(110). The resulting PMMs were corrected for the polarisation-dependence of the photoemission matrix element and symmetrized with respect to the geometrical centre. Refer to the Supplementary Methods for details.

**DFT calculations.** DFT calculations were performed to identify and characterise possible AGs and to compute the molecular orbitals required for the simulation of the ARPES data. Isolated gas-phase molecules were calculated at the PBE/cc-pVDZ level of theory as implemented in GAUSSIAN[32]. The CoPyr and Pyr on Ag(110) adsorbate systems were computed using the CP2K/QUICKSTEP simulation package[33] with implemented Gaussian and plane wave formalisms. The Ag surface was described via a slab model, periodic boundary conditions were applied. To prevent interaction between repeated slabs, 20 Å of vacuum space was added along the surface normal. Pyr and CoPyr monomers were placed on a 8 × 5 Ag(110) slab with five layers. The four topmost layers were relaxed, whereas the fifth was kept fixed at the bulk coordinates. Valence electrons were treated explicitly and their interaction with the atomic cores was described through norm-conserving Goedecker–Teter–Hutter (GTH) pseudo potentials[34]. The molecular orbitals were expanded in Gaussian type orbitals using double-ζ plus polarisation (DZVP) basis sets, which were optimised on molecular geometries[35]. The auxiliary plane wave basis used to represent the valence electron density in reciprocal space had an energy cutoff of 500 Ry. As exchange correlation functional, the spin polarised general gradient approximation by Perdew-Burke-Enzerhof (PBE)[36] was augmented with a nonlocal Vydrov-Voorhis van der Waals density functional, in its revised form[37] to account the dispersion contributions. We refer to these

calculations of the complete adsorbate system as "high-level" and the simple isolated gas-phase molecule calculations as "low level".

## Data availability

The data that support the plots within this paper and other findings of this study are available from the corresponding author upon reasonable request.

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

## Acknowledgements

Financial support by the Swiss National Science Foundation (NCCR MUST and Sinergia Project CRSII2_160801/1) and by the University of Zurich (URPP LightChEC) is gratefully acknowledged. Computing time was provided through the Partnership in Advanced Computing in Europe (PRACE) and by the Swiss National Supercomputer Centre (CSCS).

## Author contributions

P.K. carried out the experiments with support from G.Z., M.J., K.vA., K.W., V.F., M.H. and L.C. P.K. and L.C. processed and analysed the data and discussed them with M.H. and J.O. S.S. and R.A. synthesised and provided the molecules. Y.G. and M.I. performed the DFT simulations. P.K. and L.C. wrote the manuscript. All authors discussed the results and commented on the manuscript.

## Competing interests

The authors declare no competing interests.
