## [Peer Review File · Nature Communications]

REVIEWERS' COMMENTS:

Reviewer #1 (Remarks to the Author):

The manuscript by castiglione et al. is very well written and extremely clear even though the subject is quite complex and technical. The authors present photoemission data related to some monolayer porphyrin films on the Ag(110) surface. The data analysis is carried out at a very high level and the conclusions are sound. I was wondering about the possibility that some Ag atoms can be picked up by the empty porphyrins or exchanged with cobalt. did the author checked for that?

The only perplexity I have is about the relevance of this paper for a wider community. This paper deals about a very niche techniques on a model system (monolayer obtained in UHV on a Ag single crystal, without the application of a potential or presence of electrolyte) so it is not really related to electrochemistry. The authors hint at possible applications in the title, but after reading this paper no new information about HER process or activity of the catalysts is obtained. (in any case cobalt porphyrins are not really decent catalysts for HER). This consideration about the impact represents a personal opinion and i think the editor should decide if this paper is suitable for nature communication, for sure at the technical level it is excellent. In any case I think that the title should be changed and be more related to the actual information provided in the manuscript.

Reviewer #2 (Remarks to the Author):

In this paper, adsorption geometries and electronic structures of Co-pyrphyrin and pyrphyrin on Ag(110) were studied by LEED, ARPES, and DFT simulations. The photoelectron momentum maps of valence states in HOMO regions of the adsorbed molecules are obtained by ARPES measurements. The results are of high quality, and those are well simulated by DFT calculations, from which the adsorption geometries are reasonably determined. I think that the newly developed method for adsorption geometry determination employed in this paper will have substantial impact on the surface chemistry, especially for aromatic molecule adsorption systems. I believe that the paper should be published in Nature Communications in the present form.

Reviewer #3 (Remarks to the Author):

The title of the manuscript gave me an impression that the authors put emphasis on the electronic structure of hydrogen evolution catalyst. After reading the manuscript, I realized that the main point is the new approach of the orbital tomography which is the combination of LEED, DFT calculation, and ARPES. The authors do not try to uncover the mechanism of the hydrogen evolution catalyst based on the experimental results. Maybe the most important result related to the catalyst activity is the absence of Co 3d_{z²} component. However, the authors do not analyze this point further.

The main part of the present work is dedicated to the identification of several domains of different adsorption geometries beyond the conventional approach which requires single-orientational domain of molecules. As emphasized by the authors, the present approach is new and applicable to multi configuration systems. However, the existence of the three configurations AG1, AG2, AG3 is not really essential for the hydrogen evolution. I am afraid that the present work does not attract much attention from broad research fields unless the new approach can provide some relevant information for development or improvement of catalysts. Therefore, I cannot recommend publication of this manuscript in the present form.

Minor point: Several misprints in the caption of Fig. 3.

Maybe a misprint at the 6th line from the bottom, page 6 " found between ..."

Response to reviewer comments and modifications of manuscript:

NCOMMS-19-28166

“Valence electronic structure of a hydrogen evolution catalyst: orbital tomography of Co-pyrphyrin”

Changes to the manuscript

1) We changed the title to make clear that the focus of our work is the methodology, which was eventually demonstrated on a catalytic system. The new title reads as follows:

“Combined orbital tomography study of multi-configurational molecular adsorbate systems”

2) For the same reason, the abstract was also reworded. It is now clarified that the focus of the work is the study of multi-configurational systems. The new abstract reads as follows:

“Molecular reactivity is determined by the energy levels and spatial extent of the frontier orbitals. We use orbital tomography based on angle-resolved photoelectron spectroscopy to study the electronic structure of the hydrogen evolution catalyst Co-pyrphyrin. We characterize the catalyst on an Ag(110) substrate and compare it with the empty pyrphyrin ligand. Conventionally, orbital tomography is restricted to systems with one single rotational domain. In this work, we extend the method to systems with multiple rotational domains. In combination with low-energy electron diffraction and DFT simulations, we fully determine adsorption geometry and both energetics and spatial distributions of the valence electronic states. We find two states close to the Fermi level in Co-pyrphyrin with Co 3d character that are not present in the empty ligand. In addition, we identify several energetically nearly equivalent adsorption geometries that are important for the understanding of the electronic structure. The ability to disentangle and fully elucidate multi-configurational systems renders orbital tomography much more useful to study realistic catalytic systems.”

3) Figure captions were extended in line with editorial requirements so that they are self-consistent.

Response to reviewer comments

Reviewer #1:

"The manuscript by castiglione et al. is very well written and extremely clear even though the subject is quite complex and technical. The authors present photoemission data related to some monolayer porphyrin films on the Ag(110) surface. The data analysis is carried out at a very high level and the conclusions are sound. I was wondering about the possibility that some Ag atoms can be picked up by the empty porphyrins or exchanged with cobalt. did the author checked for that?"

We thank Reviewer #1 for the careful evaluation of our manuscript. Our work deals with pyrphyrin. In contrast to porphyrin, pyrphyrin consists of four fused pyridine subunits instead of pyrrol groups. We evaporated the molecules while the Ag(110) surface was held at room temperature. It was shown in previous studies, that metalation of the empty ligand from the surface or transmetalation only occurs at elevated temperatures exceeding 423 K (Rieger et al. *Phys. Chem. Lett.* **8**, 6193 (2017)). In general, transmetalation is also in favor of exchanging a noble metal atom with a less noble one, exchanging Co with Ag would be the opposite of this rule. We thus can exclude exchange of Co atoms with Ag in our experiments. We also did not find any indication for this in the XPS spectra.

The only perplexity I have is about the relevance of this paper for a wider community. This paper deals about a very niche techniques on a model system (monolayer obtained in UHV on a Ag single crystal, without the application of a potential or presence of electrolyte) so it is not really related to electrochemistry. The authors hint at possible applications in the title, but after reading this paper no new information about HER process or activity of the catalysts is obtained. (in any case cobalt porphyrins are not really decent catalysts for HER). This consideration about the impact represents a personal opinion and i think the editor should decide if this paper is suitable for nature communication, for sure at the technical level it is excellent. In any case I think that the title should be changed and be more related to the actual information provided in the manuscript."

The reviewer addresses a few important issues. Indeed, we looked at a model system in UHV since we consider this mostly a proof of concept study. Based on this work though it should be no problem to co-adsorb water for instance and study changes of the molecular levels. Importantly, there is no longer the restriction to find a particular model system with one single-orientational domain since we now demonstrate that multi-configurational systems can be studied as well.

Porphyrins are truly no good HER catalysts but the investigated Co-pyrphyrin showed some promising catalytic activity in earlier studies.

We agree with the reviewer that both original title and abstract possibly raised to many expectations in terms of information about the HER mechanism. We thus changed both to make sure that it is clear from the beginning that the focus and highlight of our work is the extension of the methodology to multi-configurational and more complex systems. We strongly believe that this is an important extension since most relevant systems are multi-configurational and could thus previously not be studied in detail.

Reviewer #2:

"In this paper, adsorption geometries and electronic structures of Co-porphyrin and porphyrin on Ag(110) were studied by LEED, ARPES, and DFT simulations. The photoelectron momentum maps of valence states in HOMO regions of the adsorbed molecules are obtained by ARPES measurements. The results are of high quality, and those are well simulated by DFT calculations, from which the adsorption geometries are reasonably determined. I think that the newly developed method for adsorption geometry determination employed in this paper will have substantial impact on the surface chemistry, especially for aromatic molecule adsorption systems. I believe that the paper should be published in Nature Communications in the present form."

We are grateful to Reviewer #2 for his very positive assessment of our work. We also believe that the described method has substantial impact because it renders the elegant method of orbital tomography much more accessible to complex adsorbates that are closer to realistic catalytic systems, for example.

Reviewer #3:

"The title of the manuscript gave me an impression that the authors put emphasis on the electronic structure of hydrogen evolution catalyst. After reading the manuscript, I realized that the main point is the new approach of the orbital tomography which is the combination of LEED, DFT calculation, and ARPES. The authors do not try to uncover the mechanism of the hydrogen evolution catalyst based on the experimental results. Maybe the most important result related to the catalyst activity is the absence of Co 3d_{z²} component. However, the authors do not analyze this point further."

We thank Reviewer #3 for the careful examination of our work. We apologize that both original title and abstract gave the impression that the focus of our work was the study of mechanistic details of a hydrogen evolution catalyst. Indeed, the highlight of our work is the extension of orbital tomography to multi-configurational systems. We adapted both title and abstract to make this clear and prevent any misunderstanding.

We believe that the actual identification of the frontier orbitals is indeed an interesting finding. We agree though that this alone does not completely unveil the mechanism of the catalytic reaction. This, however was not part of this proof of concept study.

"The main part of the present work is dedicated to the identification of several domains of different adsorption geometries beyond the conventional approach which requires single-orientational domain of molecules. As emphasized by the authors, the present approach is new and applicable to multi configuration systems. However, the existence of the three configurations AG1, AG2, AG3 is not really essential for the hydrogen evolution. I am afraid that the present work does not attract much attention from broad research fields unless the new approach can provide some relevant information for development or improvement of catalysts. Therefore, I cannot recommend publication of this manuscript in the present form."

We are glad to see that the reviewer considers the extension of orbital tomography beyond single-orientational systems the main part of our work. We hope the new title and rewritten abstract makes this now clear from the beginning.

We agree that the existence of several adsorptions geometries is no prerequisite for catalytic activity. However, many catalytic systems are indeed multi-configurational in the sense that the catalyst

molecule is adsorbed in different geometries or adsorption sites. Such systems could previously not be studied by orbital tomography due to the requirement of single orientation. With our extension, a much larger class of relevant catalytic systems can now be studied with unprecedented detail by orbital tomography. We believe that this extension is important since the method is no longer restricted to a few selected model systems of mostly of theoretical interest.

“Minor point: Several misprints in the caption of Fig. 3.

Maybe a misprint at the 6th line from the bottom, page 6 “ found between ...”

We thank the reviewer for pointing out these misprints. They were all corrected in the revised version of the manuscript.